# Hybrid DAER Based Cross-Modal Retrieval Exploiting Deep Representation Learning

**DOI:** 10.3390/e25081216

**Published:** 2023-08-16

**Authors:** Zhao Huang, Haowu Hu, Miao Su

**Affiliations:** 1Key Laboratory of Modern Teaching Technology, Ministry of Education, Xi’an 710062, China; zhaohuang@snnu.edu.cn; 2School of Computer Science, Shaanxi Normal University, Xi’an 710119, China; hu17691221525@163.com

**Keywords:** dual attention network, data augmentation, cross-modal retrieval, enhanced relation network

## Abstract

Information retrieval across multiple modes has attracted much attention from academics and practitioners. One key challenge of cross-modal retrieval is to eliminate the heterogeneous gap between different patterns. Most of the existing methods tend to jointly construct a common subspace. However, very little attention has been given to the study of the importance of different fine-grained regions of various modalities. This lack of consideration significantly influences the utilization of the extracted information of multiple modalities. Therefore, this study proposes a novel text-image cross-modal retrieval approach that constructs a dual attention network and an enhanced relation network (DAER). More specifically, the dual attention network tends to precisely extract fine-grained weight information from text and images, while the enhanced relation network is used to expand the differences between different categories of data in order to improve the computational accuracy of similarity. The comprehensive experimental results on three widely-used major datasets (i.e., Wikipedia, Pascal Sentence, and XMediaNet) show that our proposed approach is effective and superior to existing cross-modal retrieval methods.

## 1. Introduction

With the accelerated advance of multimedia information, a large number of applications require cross-modal retrieval, especially using the text queries to look for relevant images and vice versa. Cross-modal retrieval is emerging as a new technology of information retrieval, which aims to determining whether information from different modalities point to the same content [1]. Cross-modal retrieval normally involves two important stages, including data feature extraction and similarity calculation [2]. Data feature extraction is about withdrawing the vital features from data that uniquely identifies them [3], while the similarity calculation refers to computing the similarity between different modal data [4]. It is important to note that different information modalities have unbalanced and complementary relationships. For example, although the same semantic contents are described in image and text, the information that they contain can be completely different [5]. It is hard to directly measure the correlation between cross-modal information due to the distribution gap and heterogeneity.

Many information retrieval techniques have been proposed for information retrieval in recent years (e.g., [6,7]). Some focus on one-way information retrieval (e.g., using a text query to search for images) or single-modal application scenarios (e.g., within text or images), which lack sufficient attention to multi-modal information fusion (e.g., [8,9]). Some develop image and text cross-modal retrieval methods that are mainly based on shallow models to learn linear projections of image and text features (e.g., [10,11]). It can be arguable that such methods are limited for data with nonlinear subspace structure [12]. Others that investigate the nonlinear mapping of data (e.g., [13]) present difficulties in kernel selection and massive data handling.

Among these techniques, deep learning-based methods (e.g., CNN and RNN) are popularly used for cross-modal retrieval, which can not only perform linear or nonlinear mapping of data, but process a large scale of data [14]. Such methods employ a common technique that transforms the samples from altered modalities into an accepted representation space, making connections to data from different modalities in this space. Then, the similarity is calculated by using a predefined distance calculation formula, such as the cosine similarity formula [15]. However, the significant defects of the deep learning-based methods can be found in terms of the feature extraction and the similarity calculation from the cross-modal retrieval process. More specifically, regarding extracting text features, the deep learning-based methods usually assume that each word can fully express the user’s true intention with the same level of importance in the text. In fact, information from different modalities (e.g., images and text) are often unbalanced and unequal [16]. The eigenvectors between different modalities are difficult to precisely match. For similarity calculations, the deep learning-based methods also assume that images and text have equal amounts of information. Actually, it is difficult to directly use distance calculation formulae to measure the similarity between different modalities.

Thus, it is important to develop an efficient approach that can eliminate the cross-modal gap [17], increase the efficiency of information usage [18], and address users’ needs for cross-modal information retrieval [19]. To this end, this study primarily focuses on the cross-modal retrieval method of image and text. More specifically, the following research questions are investigated: *(1) How can the weights of different fine-grained size regions in text and images be precisely extracted? (2) How can the similarity between text and images be accurately calculated?* To answer these questions, in this study, a novel method for cross-modal retrieval is proposed that integrates a dual attention network and an enhanced relational network. The dual attention network is used for precisely extracting the features from images and texts, while the enhanced relational network aims to improve the computational accuracy of relational networks.

This study contributes to cross-modal information retrieval research by developing a novel image-text retrieval method. This method addresses the dual attention network and the enhanced relational network with different activation functions to fit in the dimension gap between the cross-modal information. Such a method not only minimizes the discrepancy between heterogeneous information, but keeps the nonlinear ability to learn the common representation space. Moreover, this study increases the understanding of the multiple stages of cross-modal retrieval of images and text, as well as strengthening the knowledge of feature extraction and the similarity calculation between different modalities. Finally, this study offers deeper insights into the use of attention mechanisms in the process of image and text feature extraction, and multimodal features’ similarity calculation, which are beneficial for achieving more desirable cross-modal retrieval outcomes.

The paper is structured as follows. A systematic review of cross-modal retrieval is introduced in Section 2. This is followed by describing our proposed cross-modal retrieval approach (Section 3). Section 4 reports and discusses the comprehensive experimental results. Section 5 concludes this work and points to the limitations and future research directions.

## 2. Related Work

Cross-modal information retrieval is an important information retrieval technology that attempts to find the same information from different modalities [20]. Common cross-modal retrieval methods are generally categorized into two groups: traditional cross-modal representation learning methods [21] and deep cross-modal representation learning methods [22]. The traditional methods aim to map the features of different modalities into a common space and produce a common accepted representation. Canonical correlation analysis (CCA) is one of the most representative methods, which tends to gain the common subspace of two modalities by maximizing the correlation between images and texts [22]. However, the correlation between images and texts is complicated and nonlinear, which makes it difficult to build a linear projection model [23]. Thus, CCA is inadequate when mapping the nonlinear relationship between different modalities. Some nonlinear methods are suggested to overcome the above issues. For example, Jia et al. [24] used a multi-label kernel canonical correlation analysis (ml-KCCA) to enhance the kernel CCA through the high-level semantic information reflected in multi-label annotations. Furthermore, Xu et al. [25] employed a correlation-based cross-modal subspace learning method that learns a subspace representation for each modality by maximizing the kernel dependence. However, the challenges of selecting the most appropriate kernel function still remain.

Deep cross-modal representation learning methods can perform both linear and nonlinear mapping of data without addressing the kernel function. These methods can simulate the way neurons work in the human brain and achieve great success with their robust fitting ability in various tasks, such as pattern recognition and data classification [26]. Deep learning has the advantages of distributed feature representation, automatic feature extraction and good generalization ability, which greatly alleviates the challenge of similarity measurement between different modalities in cross-modal retrieval. The cross-modal retrieval method based on deep learning can extract high-level semantic correlation on large-scale datasets and can continuously optimize the processing of multi-modal data and feature extraction problems. Several studies have developed diverse deep cross-modal representation learning methods that utilize the neural networks to extract image and text features and build relationships between different modalities. For example, Gao et al. [27] proposed an image encoder, text encoder, and multi-modal encoder to extract text features and image features and mine rich feature information. Viviana et al. [28] used convolutional neural networks to deeply extract features from images and text in the stage of establishing connections between different modal data. Similarly, Wu et al. [29] used MS^2^GAN to label information to model inter-modal and intra-modal similarities through joint learning of specific modes and shared feature representations. Although abovementioned deep cross-modal representation learning methods can perform effective cross-model retrieval, every piece of information within the cross-model data is assumed to be the equal importance. Indeed, the importance of each piece of information in the data differs from various tasks [30].

An attention network can distinguish the importance of different content in the data within a diversity of tasks [31]. Moreover, it can overcome the challenges in recurrent neural networks, such as performance degradation as input length increases, or computational inefficiency due to unreasonable input order. An attention network is originally used in natural language processing or machine translation [32]. It can also focus on certain regions in the image and assign more weights to those regions, solving the problem of treating all inputs equally in traditional neural networks [33]. Therefore, the attention mechanism has attracted much attention in natural language processing (e.g., [34]), computer vision (e.g., [35]), voice processing (e.g., [36,37]) and hash-based cross-modal retrieval of images and texts (e.g., [30]). For example, Peng et al. [38] used a visual-textual bi-attention mechanism to distinguish the visual and textual fine-grained patches as well as their relations with different saliency. Jin et al. [19] proposed a coarse-fine-grained parallel attention mechanism for video-text cross-modal retrieval, which enhances the relationship between feature points in the same modal features. Peng et al. [39] proposed a cross-modal attention block to shrink the gap between features from different modalities and focus on the relevant correlations. Dong et al. [40] used the attention mechanism to select more prominent features from images and texts, and filter out unimportant information in order to gain a more discriminative feature representation in the common latent space. Additionally, Chen et al. [41] proposed an iterative matching with recurrent attention memory (IMRAM) method that exploits the attention mechanism to explore the alignment between the features of image and text. Furthermore, Wang et al. [42] adopted a self-constraining and attention-based hashing network (SCAHN) for bit-scalable cross-modal hashing. The results show that the hash codes can be more accurately ranked by using the attention to weight the hash codes in the process of feature fusion.

For precise similarity calculation between different modalities, a relation network is importantly introduced, which can be described as a neural network structure that can entirely mine the correlation between different objects without addressing the alignment problem between the contents of the objects [14]. Sung et al. [43] proposed a two-branch relation network, which is used to determine the category of the query image by computing the distance between the query image and various types of images. Likewise, Santoro et al. [44] investigated relation networks and relational reasoning in neural networks. Although similarity calculation has been achieved by adding feature representations to relation networks, using excessive similarity relation during learning feature representation can weaken the modality-specific semantics and make unexpected noise, which influences overall performance.

## 3. The Proposed Method

To fill the gaps in cross-modal data where the original representations of image and text cannot directly measure the similarity, this study constructs a dual attention network and an enhanced relation network (DAER) to learn modality-specific features from various modalities to calculate the similarity between image and text. The overall framework of our proposed DAER cross-modal retrieval approach is presented in Figure 1. The proposed approach comprises three modules, including a text feature extraction module, an image feature extraction module, and an enhanced relation network module. Specifically, for text feature extraction module, the study uses Word2Vec and LSTM model to extract text features initially. Then, the dual attention network is used to extract the weight information of different region sizes in text. In the dual attention network, a larger pooled kernel is employed to process the weight information of features in a larger region in the text, while a smaller pooled kernel is implemented to process the weight information of features in a smaller region in the text. These two different types of weight information are passed through two multi-layer perceptions to further extract the weight information. As for the image feature extraction module, this study adds a dual spatial attention network to the bottleneck module of the MobileNetV3-large model. For the weight information of features in a larger space in the image, a larger convolution kernel is used to extract the weight information of features. Regarding the weight information in the smaller space in the image, the smaller convolution kernel is used to extract the weight information of features. Finally, the image features and text features are concatenated by the enhanced relational network module, and the concatenated data are enhanced exponentially. The similarity between images and texts is calculated by multi-layer perceptron.

### 3.1. Text Feature Extraction Module

Within the text feature extraction module, this study uses Word2Vec and LSTM [45] to initially extract the data features of the text. These features also contain the contextual relationships in the text. Furthermore, a dual region attention network is employed to analyze the importance of different fine-grained words in the text. Two pooling kernels of different sizes are processed for the representation $X_{q}^{t}$ of the *q*-th text obtained from Word2Vec and LSTM. For the small pooling kernels, the maximum pooling and the average pooling methods are used with the purposes of processing text data. The outcomes generated from these two methods are added and passed through a multi-layer perceptron. After that, the smaller fine-grained weight information can be gained by the sigmoid function (see the formula in Equation (1)). The same process is also conducted for the large pooling kernels to obtain the larger fine-grained weight representation $X_{q}^{tb}$ of the text (see the formula in Equation (2)). The representations of the text with the different fine-grained importance levels are obtained by multiplying $X_{q}^{ts}$, $X_{q}^{tb}$ and $X_{q}^{t}$ (see the formula in Equation (3)).
(1)$$X_{q}^{ts}=\sigma \left(f_{\theta }\left(W_{0}\left(F_{avg}^{s}\right)+W_{0}\left(F_{max}^{s}\right)\right)\right)$$
(2)$$X_{q}^{tb}=\sigma \left(f_{\vartheta }\left(W_{1}\left(F_{avg}^{b}\right)+W_{1}\left(F_{max}^{b}\right)\right)\right)$$
(3)$$X_{q}^{tf}=X_{q}^{ts}\;*\;X_{q}^{tb}\;*\;X_{q}^{t}$$
where $X_{q}^{ts}$ represents a smaller fine-grained weight representation of the text obtained by a smaller pooling kernel, and $X_{q}^{tb}$ represents a larger fine-grained weight representation of the text obtained by a larger pooling kernel. σ is the sigmoid activation function. $f_{\theta }$ represents a multi-layer neural network; θ represents the parameters of this neural network. $F_{avg}^{s}$ is the smaller average pooling kernel, $F_{max}^{s}$ is the smaller max pooling kernel, $F_{avg}^{b}$ is the bigger average pooling kernel, $F_{max}^{b}$ is the bigger max pooling kernel, $W_{0}$ is the shared weight matrix of the smaller pooling kernels and $W_{1}$ is the shared weight matrix of the bigger pooling kernels. $X_{q}^{t}$ represents the *q*-th text representation obtained from Word2Vec and LSTM.

The output dimension of this module is unified through the fully connected layers. This study uses the stochastic gradient descent method to perform the classification training on the text dataset, and employs the cross-entropy loss function to optimize the neural network of this module.

### 3.2. Image Feature Extraction Module

Within the image feature extraction module, the image data features are extracted based on MobileNetV3 model [46]. More significantly, this study incorporates a dual spatial attention network in the bottleneck module of MobileNetV3-large module. The structure of the improved bottleneck module is presented in Figure 2.

As showed in Figure 2, average pooling and the max pooling are conducted to obtain the pooled representation of the original data. Convolution kernels with different sizes are employed to perform the convolution operations on the polling results. The results of these two convolutions are passed through a sigmoid activation function. Finally, the original representation of the image is combined with the outputs of the spatial attention network to gain the final representation of the image. The specific formulas of these processes are described in Equations (4)–(6), respectively.
(4)$$X_{p}^{is}=\sigma \left(f^{n*n}\left(\left[F_{avg};F_{max}\right]\right)\right)$$
(5)$$X_{p}^{ib}=\sigma \left(f^{m*m}\left(\left[F_{avg};F_{max}\right]\right)\right)$$
(6)$$X_{p}^{if}=X_{p}^{is}\;*\;X_{p}^{ib}\;*\;X_{p\;}^{i}$$
where *p* represents the *p*-th image, $X_{p}^{is}$ represents the smaller fine-grained weight representation of the image obtained through the smaller convolution kernel, and $X_{p}^{ib}$ represents the larger fine-grained weight representation of the image obtained through the larger convolution kernel. σ is the sigmoid activation function, *F*_*avg* is the average pooling kernel, and *F_max* is the max pooling kernel, while *f*(.) represents the convolution operation. n is the dimension of the smaller convolution kernel and m is the dimension of the bigger convolution kernel. $X_{p}^{i}$ represents the image representation obtained from MobileNetV3 model.

### 3.3. Enhanced Relation Network

The procedure of mutual retrieval of different modal data can be considered as the classification process of distinguishing between one piece of modal data and another. The data classification is based on the special distribution of the data in a certain space. The changes of the data distribution and the growth of the discrepancy between the distribution of various data categories can make for more accurate data classification. Taking the text retrieval based on images as an example, first we obtain the data features of the image, and then pick out the category that has a similar distribution of images and texts. Therefore, the process of cross-modal retrieval is actually a process of classifying data according to the different distribution characteristics of data. The exponential function can change the original distribution state of the data according to the characteristics of the data itself. By choosing the appropriate power, the differences between the different types of data can be enlarged while ensuring the original distribution characteristics of the data.

As presented in Figure 3, normal distribution is used as an example. The original distribution of these two normal kinds of data is shown on the left, and the distribution of these two kinds of data after exponential enhancement is presented on the right. It seems clear that there are many intersections between these two kinds of data in the original distribution. However, the intersections between the two kinds of data are largely reduced after the exponential enhancement. This evidence of using the exponential functions can increase the difference between different categories of data.

To make full use of the characteristics of the extracted image and text data, we first combine the features extracted from both text and images, and this fusion mechanism is denoted as Equation (7).
(7)$$R_{qp}^{ti}=g\left(R_{q}^{t};R_{p}^{i}\right)\;$$
where $R_{q}^{t}$ represents the final representation of the *q*-th text passed through the text feature extraction module, $R_{p}^{i}$ is the final representation of the *p*-th image passed through the image feature extraction module,$R_{qp}^{ti}$ is the fused representation of the *q*-th text and the *p*-th image. $g\left(R_{q}^{t};R_{p}^{i}\right)$ represents the fusion mechanism. Then, we perform data enhancement operation on $R_{qp}^{ti}$; the specific formula is shown in Equation (8),
(8)$$R_{qp}^{tie}=pow\left(R_{qp}^{ti},k\right)$$
where $R_{qp}^{tie}$ represents the data after data augmentation, *pow*(.) represents the power operation, and k represents the power of the data. For the negative numbers in the original data, we first find their absolute value, then find the k power of the absolute value; after that, we multiply it by −1, keeping its sign unchanged. This process is as shown in Equation (9),
(9)$${R_{qp}^{ti\;\prime \prime }=-1\;*\;pow(abs(R}_{qp}^{ti\;\prime }),\;k)$$
where $R_{qp}^{ti\;\prime }$ are the negative values in $R_{qp}^{ti}$, and abs(·) represents the operation of getting the absolute value of $R_{qp}^{ti\;\prime }$.

Finally, we use several layers of neural networks to calculate the similarity of the paired image-text data. The specific formula is described in Equation (10),
(10)$$S_{qp}^{ti}=R\left(R_{qp}^{tie};\theta_{r}\right)$$
where $S_{qp}^{ti}$ represents the similarity between the *q*-th text and the *p*-th image. R(.) represents the neural networks and $\theta_{r}$ represents their parameters.

Following [17], we use the following Equation (11) as the objective function to optimize the relational network:(11)$$Loss={\left|\left|T-P\right|\right|}_{F}^{2}\;$$
where *T* represents the actual image-text similarity matrix, and *P* represents the image-text similarity matrix predicted by the model. ${\left||\cdot |\right|}_{F}$ is the Frobenius norm.

### 3.4. Process of Model Training

In the process of text-image cross-modal retrieval, it is important to perform model training. The overall procedure of DAER is indicated in Algorithm 1.**Algorithm 1:** The training process of DAER.**Input:** The image data $X_{i}$, the text data $X_{t}$, the corresponding class label set $Y_{i}$ and $Y_{t}$

**Output:** The optimized DAER model.The training phase:
Step 1:Train the text feature extraction module:Make it perform text classification training and save its model-specific parameters.Step 2:Train the image feature extraction module:Make it perform image classification training and save its model-specific parameters.Step 3:Train the relation network:While not converging the following:Step 3.1: Extract text features using the initially trained image feature extraction module;Step 3.2: Extract image features using the initially trained image feature extraction module;Step 3.3: Fuse the image and text representations according to Equation (7) as:
$R_{qp}^{ti}=g\left(R_{q}^{t};R_{p}^{i}\right)$
   Step 3.4: Perform data enhancement on the fused data according to Equation (8) as:$R_{qp}^{tie}=pow\left(R_{qp}^{ti},k\right)$
   Step 3.5: Use Equation (11) as the objective function to optimize the relationnetwork and adjust the parameters of image and text feature extractionmodules by stochastic gradient descent.$Loss={\left|\left|T-P\right|\right|}_{F}^{2}$
Step 4: End while.Step 5: Save the parameters of the overall model.

## 4. Experimental Work

The experimental work details the experiment design to validate our proposed DAER model. It covers the datasets, evaluation metrics, comparison methods, and experiment implementation.

### 4.1. Datasets

Three standard benchmark datasets were selected and used to experimentally verify the capabilities of our proposed DAER model in this study. They are Wikipedia, Pascal Sentence, and XmediaNet. The data partition scheme in this study follows the setting in [43], in which each dataset is split into three subsets, covering 80% of pairs for training, 10% pairs for verification, and 10% pairs for testing. Wikipedia is the most widely used dataset for measuring the performance of the cross-model retrieval method. It is collected from “the featured articles” and composed of 2866 image-text pairs, which are grouped into 10 popular categories [11]. Each image relates to a complete text article. Similar to the data partition scheme of [43], we also grouped the database into three subsets, including 2292 pairs for training, 286 pairs for verification, and 287 pairs for testing. Pascal Sentence consisted of 1000 image-text pairs, which can be categorized into 20 semantic groups [13]. Each image was produced from the 2008 PASCAL development kit, and their corresponding text samples were obtained by distinct annotators of the Amazon Mechanical Turk. There were 800 pairs for training, 100 pairs for verification, and 100 pairs for testing. XmediaNet is a large-scale dataset, which covers five major sections, containing 40,000 texts, 40,000 images, 10,000 audio files, 10,000 videos, and 2000 3D models. Each section has about 200 classes [17]. This study exploited images and texts only to evaluate the effectiveness of the proposed methods. Thus, we finally obtained 40,000 image-text pairs. The training set had 32,000 pairs, the validation set had 4000 pairs, and the testing set had 4000 pairs.

### 4.2. Evaluation Metric

To assess the performance of cross-modal retrieval, two types of cross-modal retrieval tasks were performed on the selected datasets. One type of task was text retrieval based on the given image (image to text), another was image retrieval based on the given text (text to image). Mean average precision (mAP) was employed as the evaluation metric for our proposed approach, as mAP is the mean value of average precision (AP) scores for each query. The definition of AP is given as Equation (12):(12)$$AP=\frac{1}{R}\sum_{k=1}^{n}\frac{R_{k}}{k}\;*\;{rel}_{k}$$
where *n* is the number of total instances, and *R* is the number of relevant instances. $R_{k}$ is the number of relevant instances in the top *k* returned results. ${rel}_{k}$ is set to be 1 when the *k*-th returned result is relevant. Otherwise, ${rel}_{k}$ is set to be 0.

### 4.3. Comparison Methods

To validate the effectiveness of our proposed DAER model, eight existing image-text cross-modal retrieval methods were selected in our comparative experiments. These included four traditional cross-modal retrieval methods, namely CCA [22], KCCA [23], MvDA-VC [47], and JRL [48], and four deep learning-based methods, namely ACMR [16], CM-GANs [49], FGCrossNet [18], and MCSM [4].

Regarding the traditional cross-modal retrieval methods, CCA learns projection matrices to map the features of different modalities into a common space by maximizing the correlations between images and texts [22]. KCCA adopts the kernel function to extend CCA for common space learning. In these experiments, the Gaussian kernel was employed as the kernel function [23]. MvDA-VC optimizes the generalized Rayleigh quotient, which can maximize the inter-class variation and minimize the intra- and inter-view variation within a common space [47]. JRL uses semi-supervised regularization and dispersed regularization to apprentice the accepted amplitude with semantic information [48]. Note that the key distinction between the above traditional methods and our proposed DAER is that our approach focuses on the deep learning method that addresses the dual attention networks to extract features of the image and text, giving different weights to different information in the extracted features.

Deep learning-based methods usually use a predefined similarity calculation formula to directly process the image and text data. Specifically, ACMR finds common spaces between data of different modalities by exploiting adversarial networks [16]. CM-GANs leverage adversarial networks to generate common representations between modalities, thereby bridging the heterogeneity gap between modalities [49]. FGCrossNet is a fine-grained cross-modal retrieval method, which addresses three constraints in the common space, including the centrality constraint, the classification constraint, and the ranking constraint [18]. MCSM uses a recurrent attention network with an attention-based joint embedding loss to model the specific features within each modality [4]. It is important to note that having extracted the features of images and texts with different fine-grained weight information in this study, the data of images and texts were augmented by keeping the original distribution of data and expanding the differences between different categories of data. After that, the relationship network was employed to compute the similarity between the extracted features.

Moreover, to further explore the validity of our proposed DAER model and gain deep insights into the specific impacts of our proposed model, four variants of the DAER were developed and comparatively evaluated in this study. These variants were DAER-C (i.e., no improvement and using as a blank control test), DAER-I (i.e., only adding improvements to the image feature extraction module), DAER-T (i.e., only adding improvements to the text feature extraction module), and DAER-I-T (i.e., adding improvements to both text and image feature extraction modules).

### 4.4. Experiment Implementation

In the experiments, we followed the feature exaction strategies. For the text feature extraction, the representation dimension of each word was set as 300, and the number of neurons in the LSTM was set as 300. In the dual region attention module, the size of the small pooling kernel was set as 3 and the size of the large pooling kernel was set as 7. The number of the hidden layers at the excitation stage was set as 3, and the numbers of the neurons on each layer were 512, 1024, and 512, respectively. For a single text, the final output dimension of this module was 1 × 300.

To extract the image feature, the size of each image was initially changed to 224 × 224 pixels, and the size of the smaller convolution kernel was set as 3 × 3 in the dual spatial attention module. We also set the size of the larger convolution kernel as 7 × 7. Finally, the classification layer was removed from the MobileNetV3-large structure after initial training completion. For a single image datum, the final output dimension of this module was 1 × 300.

For integrating image and text in the relational network, we adopted the fusion method of splicing [17]. We set 0.6 as the most appropriate number of powers in data augmentation. The relational network was implemented by a four-layer neural network, and the numbers of the neurons on each layer were 600, 1024, 512, and 1, respectively.

The proposed DAER was implemented with the Tensorflow2.4, and the experimental work was conducted on a PC with a i9-10900k CPU, 64G, NVIDIA Quadro RTX4000 8G, and Windows 10. Figure 4 presents the examples of image and text queries using the Wikipedia dataset.

## 5. Results and Discussion

### 5.1. Comparison with Existing Methods

To verify the effectiveness of our proposed DAER approach, we compared our approach with eight existing cross-modal retrieval methods in the experiments, including four traditional cross-modal retrieval methods, namely CCA [22], KCCA [23], MvDA-VC [47], JRL [48], and four deep learning-based methods, covering ACMR [16], CM-GANs [49], FGCrossNet [18], and MCSM [4]. Cross-modal retrieval accuracy was used to evaluate the effectiveness of the learned common representation of both our proposed approach and the compared methods. Table 1 shows the comparative results of three target datasets, namely Wikipedia, Pascal Sentence, and XMediaNet. Overall, the results show that our proposed DAER had the highest scores among the comparative methods in both retrieval tasks (i.e., image-to-text retrieval and text-to-image retrieval) within three target datasets. Furthermore, it can be found that the our proposed DAER approach achieved the greatest scores of average accuracy in three target datasets, which were around 0.502, 0.689, and 0.691, respectively. These findings suggest that our proposed DAER approach had the best cross-modal retrieval performance among the selected methods. This may be because the dual attention network could accurately extract the weight information of image and text data in different fine-grained spaces. Additionally, the enhanced relation network could expand the difference between different modalities while maintaining the original distribution characteristics, so as to achieve a more accurate similarity calculation between different types of data. 

To further verify the superiority of our proposed DAER approach, statistical tests were also used to assess the differences between different search methods. This helped determine if there were statistically significant differences in performance between different methods. This study used the Wilcoxon signal-rank test [50] with a significance level of 95% and the Bonferroni correction factor to control the error rate of multiple comparisons. As shown in Table 2, almost all *p*-values were less than 0.006 (0.05/8), which may imply that the performance difference between DAER and each comparison method was statistically significant. However, the *p*-value of method ACMR was slightly greater than 0.006, which may be because the same difference value existed in the MAP results of 10 random running instances, making it difficult to perform an exact *p*-value calculation.

Furthermore, the comparison results of the average retrieval accuracy show that the deep learning-based methods had better performance than the traditional methods in three target datasets. Such results may further confirm the superiority of the deep learning methods for feature extraction. Additionally, the findings indicate that CCA had the lowest retrieval accuracy among the four traditional cross-modal retrieval methods, which may imply that the linear mapping of data cannot sufficiently extract the features of data. This is also reflected by previous studies, showing that using a nonlinear model for data feature extraction can be more effective than using a linear model [48]. Interestingly, the findings show that JRL that utilizes a semi-supervised sparse regularization approach achieves greater retrieval accuracy than the deep learning-based methods, such as ACMR and CM-GANs. This may suggest that the traditional methods are not always inferior to deep learning-based methods in cross-modal retrieval accuracy. As indicated by Xu et al. [17], JRL can outperform multi-modal DBNs in cross-modal search.

To further illustrate the efficiency of our proposed method, the comparison results of the average retrieval accuracy on each dataset are presented in Figure 5. As shown in the figure, our proposed DAER achieved the most significant improvements for both retrieval tasks on the XMediaNet dataset, showing that the mAP values of our proposed DAER in each task were improved by 8.45%, 7.82%, and 8.14%, respectively. This was followed by the Pascal Sentence dataset, and the Wikipedia dataset placed last. Such results point out the adaptability of our proposed DAER for the large dataset. As described, the XMediaNet dataset was the largest dataset among three target datasets, containing 40,000 image-text pairs, while the Wikipedia and Pascal Sentence datasets only covered 2866 and 1000 image-text pairs, respectively.

Furthermore, our proposed DAER had greater improvements than FGCrossNet in the Pascal Sentence dataset, showing that the mAP values of our approach on both retrieval tasks were improved by 7.06%, 5.14%, and 6.0%, respectively. A possible explanation may be that the text in Pascal Sentence is generated by the corresponding images, which makes the content of the text more concentrated, and the relationship between the text and its corresponding image closer. Thus, there is no big difference between each model in text feature extraction. For example, the mAP value of FGCrossNet was only 0.034 lower than that of DAER in the task of retrieving images by text. Our proposed model improved the average mAP value from 0.650 to 0.689 due to the use of the image dual spatial attention mechanism and the enhanced relation network. The former can accurately extract the importance of different fine-grained spaces in the image, while the latter can make full use of the characteristics of the data themselves to expand the gaps between different types of data. By doing so, it can not only improve the accuracy of similarity calculation between different modalities, but avoid the reduction of accuracy that is caused by the data alignment issue between different modalities.

Figure 6 presents the performance of two retrieval tasks (i.e., image to text, and text to image) using different methods in the Wikipedia, Pascal Sentence, and XMediaNet datasets, respectively. Overall, it seems clear that our proposed DAER achieved greater mAP values for both retrieval tasks in each dataset. These findings further support our previous results, showing that our proposed approach achieved the best performance among the selected cross-modal retrieval methods.

Furthermore, the results show that there was a big gap between the two mAP values of each selected method in the Wikipedia dataset. In particular, the mAP value of MCSM on the image-to-text task was 12.7% higher than that on the text-to-image task, and the mAP value of CM-GANs on the image to text task was 11.8% higher than that on the text-to-image task. This may be because the content of each text was rich and complex in the Wikipedia dataset, and the text contained information from two or more categories at the same time, resulting in relatively small gaps between texts in different categories. Therefore, it can be hard to correctly extract effective information when extracting text features in Wikipedia.

### 5.2. Comparison with Variants of DAER

To measure the effectiveness of our proposed DAER approach, our DAER was compared with four variants among three datasets, including DAER-C (i.e., no improvement and using as a blank control test), DAER-I (i.e., only adding improvements to the image feature extraction module), DAER-T (i.e., only adding improvements to the text feature extraction module), and DAER-I-T (i.e., adding improvements to both text and image feature extraction modules). The comparison results are summarized in Table 3.

As shown in Table 3, the average mAP values of DAER-C, which used the blank control test, had the lowest scores among the three datasets, which were around 0.487, 0.665, and 0.657, respectively. In contrast, the average mAP values of DAER-I, which added the dual spatial attention mechanism to the image feature extraction, had higher scores, which reached about 0.492, 0.675, and 0.669 in the three datasets, respectively. This may imply that the incorporation of the dual-region attention network into the text feature extraction was helpful in extracting the importance of the different phrases and sentences at the different fine-grained levels. Similarly, the average mAP values of DAER-T, which applied the dual region attention mechanism to the text feature extraction, reached around 0.495, 0.672, and 0.674 in the three datasets, respectively, which were also greater than those of DAER-C. Such results may suggest that the addition of the dual spatial attention network to the image feature extraction can better extract the importance of information in different sizes of spatial locations for the entire image. Furthermore, the average mAP values of DAER-I-T had the best scores among the four variants, which were about 0.497, 0.681, and 0.682 in the Wikipedia, Pascal Sentence, and XMediaNet datasets, respectively. This may suggest that the combination of the two dual attention mechanisms achieved greater impacts. More significantly, compared with DAER-I-T, our DAER had better average mAP scores in the three datasets, which reached about 0.501, 0.689, and 0.691, respectively. This may be because our approach not only used a dual-region attention network in the text feature extraction and a dual spatial attention network in the image feature extraction, but also used an enhanced relation network to calculate the similarity between different modalities. By doing so, it increased the variation between data while preserving the original distribution of the data, which allowed more accurate similarity calculation between different modalities.

Figure 7 shows the different improvements of our proposed approach for the textual dual-region attention mechanism, the image dual-space attention mechanism, and the enhanced relational network used in the three databases. As shown in Figure 7, the text dual- region attention mechanism showed the most improvement in the Wikipedia and XMediaNet datasets, with average mAP values of 47% and 45%, respectively. In contrast, the improvements of the average mAP values of the image dual spatial attention mechanism in Wikipedia and XMediaNet were about 27% and 32%, respectively. These results may imply that the dual regional attention mechanism was more accurate in extracting information about the weight of different paragraphs and phrases in the text. Another possible explanation may be that our proposed dual spatial attention mechanism was adopted based on the MobileNet V3-large model with the channel attention network. This may have affected the retrieval accuracy performance of the dual spatial attention mechanism.

Interestingly, the image dual spatial attention mechanism caused the most improvement in the overall performance of our approach in Pascal Sentence, with an average mAP value of 40%, which was higher than the average mAP value of the text dual region attention mechanism. This may be because the text corresponding to the image was generated based on the image itself in Pascal Sentence, where the content itself was more concise, focused and closer to the description of the image. Therefore, the effects of the dual regional attention mechanism on text feature extraction were not particularly significant in Pascal Sentence.

Furthermore, the improvements of the enhanced relation network to the overall performance of our proposed approach were around 27%, 32%, and 29% in Wikipedia, Pascal Sentence, and XMediaNet, respectively. These findings suggest that using the characteristics of the data to exponentially augment the data themselves can increase the variation between different categories of data while maintaining the characteristics of the data distribution. Thus, the accuracy of the similarity calculation of different modal data can be significantly enhanced.

## 6. Conclusions

The study of cross-modal retrieval has increasingly attracted attention from academics and practitioners. One challenging task of cross-modal retrieval is to calculate the similarity between heterogeneous data. Current cross-modal retrieval approaches tend to jointly construct a common subspace, whereas these methods fail to consider the importance of different fine-grained information in the data, and ignore the entire utilization of the extracted data features. To overcome these challenges, this paper proposes a combination method that integrates a dual attention network and an enhanced relation network for cross-modal information retrieval. To carry out the study, a dual-region attention network was used for the text feature extraction, which attempted to extract the importance of different words and sentences at different fine-grained levels. Then, a dual space attention network was employed for the image feature extraction in order to obtain different important pieces of information of various spatial sizes. Third, according to the characteristics of the data, our study used an enhanced relation network to exponentially expand the fused image and text data, since an enhanced relation network can appropriately increase the difference in the distribution of different categories of data without considering the problem of data alignment between different modalities.

Our comparative results show that our proposed DAER had a better retrieval accuracy performance than other existing methods, demonstrating the effectiveness of our proposed approach for cross-model retrieval. Additionally, our findings confirm that the integration of a dual region attention network to the text feature extraction is beneficial in extracting the importance of different phrases and sentences at different fine-grained levels. Likewise, using a dual spatial attention network in image feature extraction is helpful to extract the importance of information in different sizes of spatial locations for the entire image. Our results also show that calculations of the similarity between the different modalities can be enhanced by using an enhanced relation network, since an enhanced relation network can appropriately increase the difference between the data and retain the original distribution of the data.

This study also has some limitations. First, this study selected three datasets to experimentally validate our proposed method, which explains the exploratory nature of our work. To give a more comprehensive assessment to draw persuasive conclusions, further study could be carried out on a wider range of datasets. Another limitation is that this study only considered text-image cross-model retrieval. It would be more interesting to extend our proposed approach to other modalities, such as video and audio, to explore the effectiveness of our approach. This study is the first step and future studies can be conducted. For example, we will focus on different perspectives of enhancing text-image cross-model retrieval performance, such as integrating generative adversarial networks, optimizing the constraints between different models in common representation space, or developing prototypes for practical validation. The findings would be valuable for developing more intelligent and effective approaches for cross-model retrieval. In addition, more accurate semantic features can be found through further supplementary prediction of semantic tags, so as to achieve more efficient cross-modal retrieval.

## Figures and Tables

**Figure 1 entropy-25-01216-f001:**
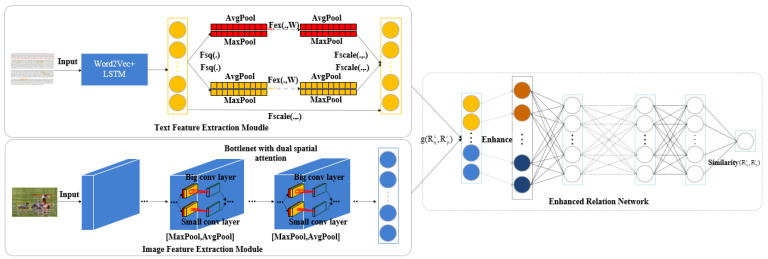
The framework of the DAER approach for cross-modal retrieval.

**Figure 2 entropy-25-01216-f002:**
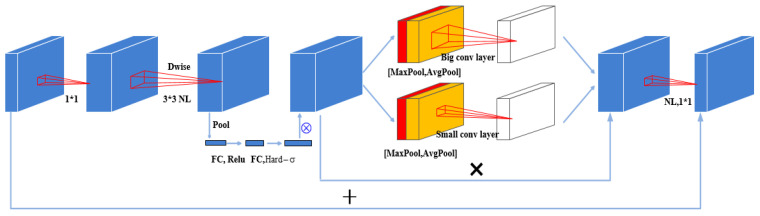
The structure of bottleneck module with dual spatial attention network.

**Figure 3 entropy-25-01216-f003:**
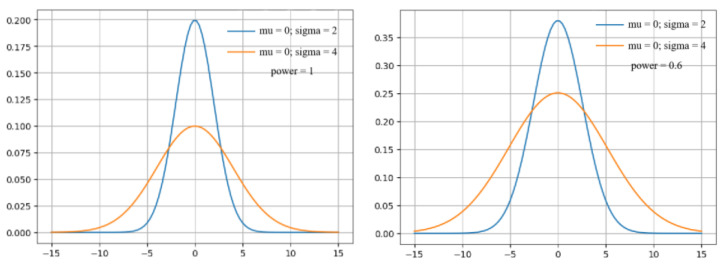
Normal distribution with two different powers (**left**: power = 1; **right**: power = 0.6).

**Figure 4 entropy-25-01216-f004:**
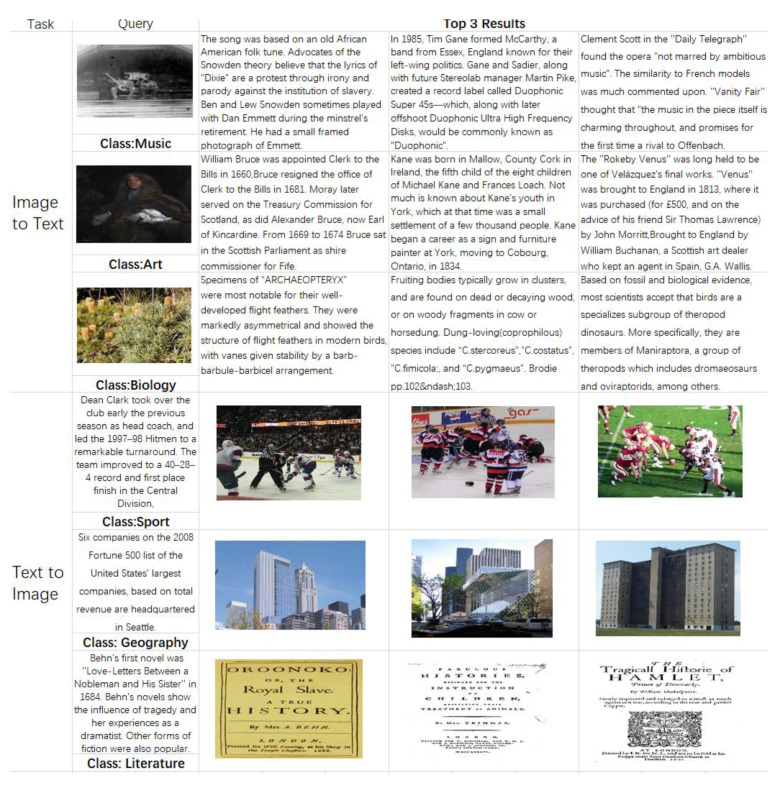
Example of retrieval tasks using the proposed DAER.

**Figure 5 entropy-25-01216-f005:**
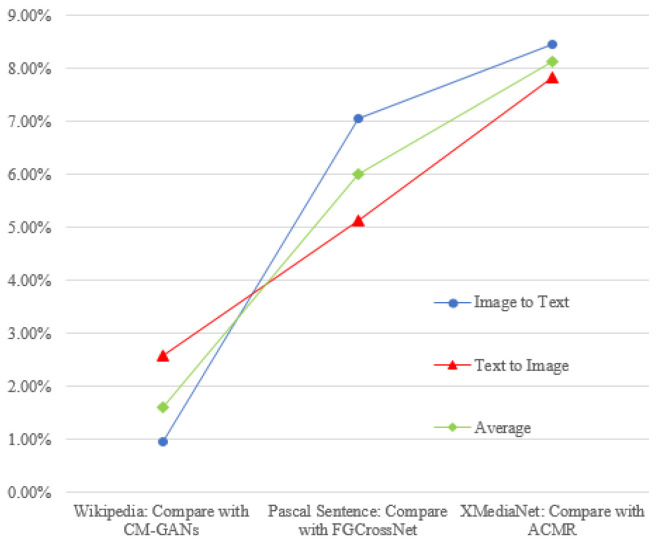
Comparison of DAER with the selected methods in three datasets.

**Figure 6 entropy-25-01216-f006:**
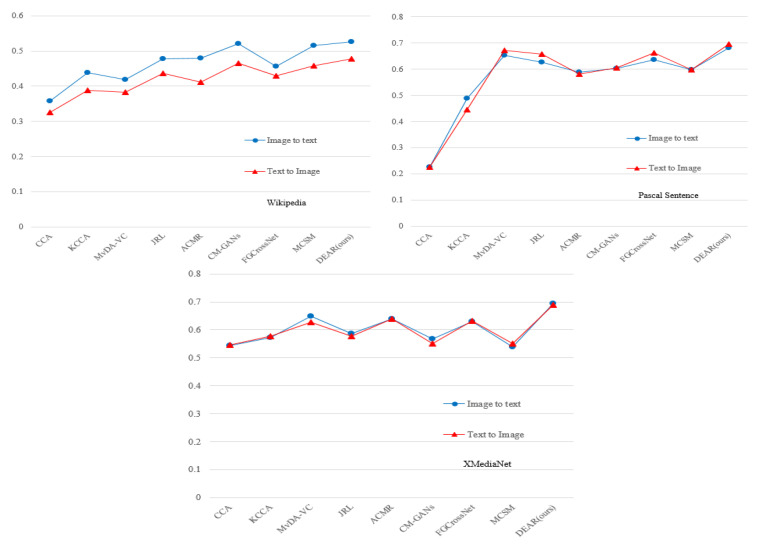
Comparison of mAP values of each method used in two tasks in three datasets.

**Figure 7 entropy-25-01216-f007:**
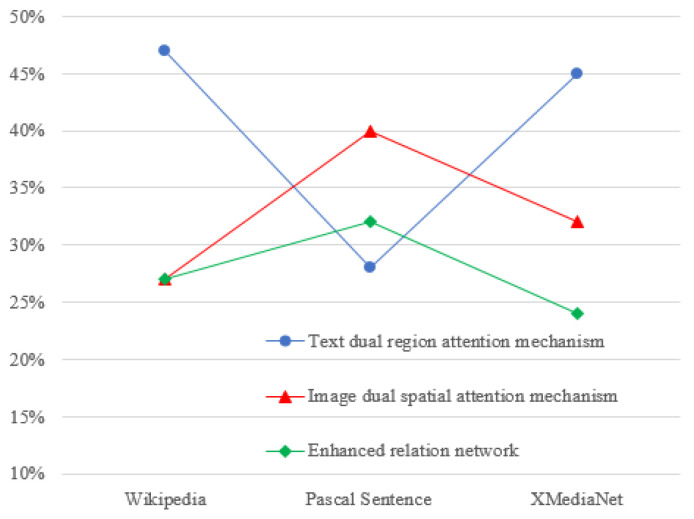
The improvement of our proposed approach in three databases.

**Table 1 entropy-25-01216-t001:** Accuracy comparison of mAP in three datasets.

Method	Image to Text			Text to Image			Average		
	D1	D2	D3	D1	D2	D3	D1	D2	D3
CCA [22]	0.357	0.225	0.544	0.326	0.227	0.546	0.341	0.226	0.545
KCCA [23]	0.438	0.488	0.573	0.389	0.446	0.577	0.414	0.467	0.575
MvDA-VC [46]	0. 419	0.652	0.650	0.382	0.672	0.627	0.401	0.662	0.638
JRL [47]	0.478	0.627	0.586	0.436	0.658	0.578	0.457	0.642	0.582
ACMR [16]	0.480	0.589	0.639	0.411	0.582	0.639	0.431	0.586	0.639
CM-GANs [29]	0.521	0.603	0.567	0.466	0.604	0.551	0.494	0.604	0.559
FGCrossNet [48]	0.457	0.637	0.629	0.429	0.662	0.633	0.443	0.650	0.631
MCSM [4]	0.516	0.598	0.540	0.458	0.598	0.550	0.487	0.598	0.545
**DAER (ours)**	**0.526**	**0.682**	**0.693**	**0.478**	**0.696**	**0.689**	**0.502**	**0.689**	**0.691**

Note: The highest score is indicated in boldface; D1 = Wikipedia; D2 = Pascal Sentence; D3 = XMediaNet.

**Table 2 entropy-25-01216-t002:** Statistical test results comparing DAER and other methods in three datasets.

Method	Image to Text			Text to Image		
	D1	D2	D3	D1	D2	D3
CCA [22]	0.002	0.002	0.002	0.002	0.002	0.002
KCCA [23]	0.002	0.002	0.002	0.002	0.002	0.002
MvDA-VC [46]	0.002	0.002	0.002	0.002	0.002	0.002
JRL [47]	0.002	0.002	0.002	0.002	0.002	0.002
ACMR [16]	0.0076	0.002	0.002	0.002	0.002	0.002
CM-GANs [29]	0.002	0.002	0.002	0.002	0.002	0.002
FGCrossNet [48]	0.002	0.002	0.002	0.002	0.002	0.002
MCSM [4]	0.0059	0.002	0.002	0.002	0.002	0.002

Note: D1 = Wikipedia; D2 = Pascal Sentence; D3 = XMediaNet.

**Table 3 entropy-25-01216-t003:** Comparison results of mAP in the Wikipedia, Pascal Sentence, and XmediaNet datasets.

Method	Image to Text			Text to Image			Average		
	D1	D2	D3	D1	D2	D3	D1	D2	D3
DAER-C	0.511	0.661	0.663	0.464	0.668	0.652	0.487	0.665	0.657
DAER-I	0.517	0.667	0.674	0.467	0.682	0.665	0.492	0.675	0.669
DAER-T	0.519	0.665	0.679	0.470	0.678	0.668	0.495	0.672	0.674
DAER-I-T	0.521	0.673	0.685	0.473	0.689	0.679	0.497	0.681	0.682
**DAER**	**0.525**	**0.682**	**0.693**	**0.478**	**0.696**	**0.689**	**0.501**	**0.689**	**0.691**

Note: The highest score is shown in boldface; D1 = Wikipedia; D2 = Pascal Sentence; D3 = XMediaNet.

## Data Availability

Data are available when required.

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
