# Peer review of "Hybrid DAER Based Cross-Modal Retrieval Exploiting Deep Representation Learning"

_entropy, 2023, doi:10.3390/e25081216_

Round 1
Reviewer 1 Report
In this paper, the authors propose a combination method that integrates the dual attention network and the enhanced relation network for cross-modal information retrieval. The work described is interesting. Experiments demonstrate good performance. However, the explanation of some figures and equations are rather brief and need more clear description. The text on the figures is too small to read. The format and language also needs improvement.
Author Response
Response: Thank you for the reviewer’ comment. Based on the reviewer’s comments, we have made the following revision
(1) We have added relevant explanation of some figures and equations [see pages 4-7].
(2) We have revised the description of some figures and equations [see pages 4-7].
(3) We have enlarged the figures in the revised manuscript [see pages 4-5].
(4) The format and language have been further improved in the revised version [see pages 1-17].
Reviewer 2 Report
This paper presents a new cross-modal retrieval model that incorporates a dual attention module and enhanced relation network. The authors have performed experiments on three datasets to validate the effectiveness of the proposed method.
While the authors argue that existing cross-modal retrieval methods do not distinguish the different importance of different window sizes in feature extractions and that the direct distance match between image and text features is inappropriate, I did not see any new techniques or insights on model designing from the proposed method. In fact, feature extractions from different window sizes and different attention weights to words and image regions are already quite common in existing work.
The paper has a few key problems. Firstly, the equations are all ill-formatted, and the images are too vague. Secondly, the proposed method has not been clearly presented, and it cannot well support the claims raised in the introduction. I did not see the dual attention module, and the techniques used for text and visual extractions are somewhat outdated. Thirdly, the equations are lacking explanations, making it hard to understand the underlying rationale. For example, the function g(.) in eq.7 and T, P are not explained. Finally, the baselines are outdated and not the state-of-the-art methods, such as the four deep learning methods before 2020.
Overall, while the paper attempts to introduce a new cross-modal retrieval model, it falls short in terms of novelty and clarity. The authors should consider revising their approach and addressing the aforementioned issues before submitting it for publication.
Author Response
The paper has a few key problems. Firstly, the equations are all ill-formatted, and the images are too vague.
Response: Based on the reviewer’ comment, we have checked and changed the format of all equations in the revised manuscript [see pages 5-8].
Secondly, the proposed method has not been clearly presented, and it cannot well support the claims raised in the introduction. I did not see the dual attention module, and the techniques used for text and visual extractions are somewhat outdated.
Response: Based on the reviewer’s comment, we have revised the description of the proposed method in the revised manuscript [see pages 4-8]
Thirdly, the equations are lacking explanations, making it hard to understand the underlying rationale. For example, the function g(.) in eq.7 and T, P are not explained.
Response: Based on the reviewer’s comment, we have further explained the equations in the revised manuscript [see pages 5-7].
Finally, the baselines are outdated and not the state-of-the-art methods, such as the four deep learning methods before 2020.
Response: Thank you for the comment. We have described that eight existing image-text cross-modal retrieval methods are relevantly selected in the study, including four traditional cross-modal retrieval methods, namely CCA, KCCA, MvDA-VC and JRL, and four deep learning based methods, namely ACMR, CM- GANs, FGCrossNet and MCSM [see page 9].
Overall, while the paper attempts to introduce a new cross-modal retrieval model, it falls short in terms of novelty and clarity. The authors should consider revising their approach and addressing the aforementioned issues before submitting it for publication.
Response: Based on the reviewer’s comment, we have revised the approach in the revised manuscript [see pages 4-9].
Reviewer 3 Report
1. The authors have carried out research on "Hybrid DAER Based Cross-modal Retrieval Exploiting Deep Representation Learning" which is interesting. But, I have certain observations.
2. The introduction is good, but, there are many intelligent techniques available in the literature. Why the authors are selecting deep learning is not clear. The motivation is missing.
3. The proposed model is presented. And then the result analysis is presented. Instead, authors could show, how one iteration work. It could help several authors to foster their research.
4. The limitation of the current research must be identified and the future scope of the research also could be presented.
Author Response
The introduction is good, but there are many intelligent techniques available in the literature. Why the authors are selecting deep learning is not clear. The motivation is missing.
Response: Based on the reviewer’s comment, we have revised the reasons choosing deep learning in the introduction. Moreover, deep learning has been reviewed and introduced in section of relevant work [see pages 2-3].
The proposed model is presented. And then the result analysis is presented. Instead, authors could show, how one iteration work. It could help several authors to foster their research.
Response: Based on the reviewer’s comment, we have added relevant information about how one iteration works in the revised manuscript [see page 4].
The limitation of the current research must be identified and the future scope of the research also could be presented.
Response: Based on the reviewer’s comment, we have revised the limitation and future research in the revised manuscript [see page 15].